# Penalty kick or not? Differences in the interpretation of handball incidents in professional association football

Tobias Bauch[1,2*], Daniel Leyhr[1,3], David Schmidt[2], Daniel Brinkmann[4], Oliver Höner[1,5]

**1** Institute of Sports Science, Eberhard Karls University, Tübingen, Germany, **2** DFB Schiri GmbH, Frankfurt am Main, Germany, **3** Methods Center, Eberhard Karls University, Tübingen, Germany, **4** DFB-Akademie, DFB GmbH & Co. KG, Frankfurt am Main, Germany, **5** Bund Deutscher Fußball-Lehrer e.V., Frankfurt am Main, Germany

* tobias.bauch@dfb.de

## Abstract

Handball decisions in the penalty area remain one of the most controversial topics in professional association football, yet they are underexplored in sports science. The purpose of this research was to establish a foundation for understanding the controversy by examining its underlying causes. Two video-based studies quantified how key stakeholders interpret handball incidents and how closely these interpretations align with Union of European Football Associations (UEFA) guidelines. Study 1 involved referees active in German men's professional football ($n = 154$) who judged 30 incidents. Study 2 repeated the procedure with professional coaches ($n = 31$) and players ($n = 46$) using 18 incidents. Outcomes were Accuracy (accordance with UEFA), Strictness (percentage of incidents deemed punishable), Consensus (within- and between-group agreement), and Reasoning (primary reason for a decision). Referees reached 84.0% Accuracy, which differed by role, performance level, and handball category. Strictness among referees was lower compared to UEFA (42.9% vs. 50%). Coaches and players demonstrated lower Accuracy (63.8% and 67.5%) and Strictness (36.0% and 33.9%) than referees, resulting in significant differences in Consensus across Stakeholder Groups in 11 out of 18 incidents. Reasoning also diverged as referees preferred Naturalness, whereas coaches and players emphasised Avoidability and Impact. These findings reveal systematic differences between governing body guidelines, referee decision-making, and practitioner expectations. The results can inform educational measures and discussions on potential revisions to the handball law. Clearer and more objective criteria, jointly agreed by key stakeholders, are required to improve the consistency and acceptance of handball decisions.

**Data availability statement:** All relevant data are within the paper and its Supporting Information files.

**Funding:** The author(s) received no specific funding for this work.

**Competing interests:** The authors have declared that no competing interests exist. Some of the authors are employed by the German Football Association (DFB). However, this affiliation did not influence the research process or outcomes. The study was designed and conducted independently, and none of the potential results would have conferred specific advantages or disadvantages to the authors or the DFB.

## Introduction

The interpretation of handball incidents remains one of the most controversial topics in professional association football, yet it is underexamined in sports science. Handball decisions frequently spark heated debates in the media and among key stakeholders, such as referees, coaches, and players. This controversy was exemplified during the EURO 2024 quarter-final between Spain and Germany. In the 106[th] minute, with the score tied at 1–1, a powerful shot by German forward Jamal Musiala was blocked by the arm of Spanish defender Marc Cucurella. The handball within Spain's penalty area possibly prevented Germany from taking the lead, but referee Anthony Taylor allowed play to continue, deciding that the incident was not punishable. The Video Assistant Referee (VAR) also did not intervene in the on-field decision. With Germany subsequently losing the match and Spain later becoming the EURO champions, the handball incident sparked significant debate. Several weeks after the tournament, the Union of European Football Associations (UEFA) Referees Committee announced that awarding a penalty kick for Germany would have been the accurate decision [1].

This example underscores the profound impact of handball decisions in professional football, especially when handball incidents occur in the penalty area. According to the Laws of the Game issued by the International Football Association Board (IFAB), a punishable handball by a defending player within their penalty area leads to a penalty kick, regardless of the likelihood of the attacking team scoring a goal in that situation [2]. Penalty kicks represent one of the most effective goal-scoring opportunities, with a conversion probability of approximately .76 expected goals (xG) [3]. In the low-scoring game of football, this frequently makes penalty kicks an essential factor for match outcomes [4]. Consequently, accurate and consistent interpretations of handball incidents in the penalty area are essential to ensure fair and equitable competition.

The ambiguity of the handball law itself contributes to this ongoing controversy. According to the Laws of the Game, a handball is considered punishable if a player "deliberately touches the ball with their hand/arm, for example, moving the hand/arm towards the ball" or "touches the ball with their hand/arm when it has made their body unnaturally bigger" [2]. The law further clarifies that a player's body is made "unnaturally bigger" if "the position of their hand/arm is not a consequence of, or justifiable by, the player's body movement for that specific situation" [2]. This wording leaves room for interpretation, and the absence of objective criteria may lead to different interpretations of the same handball incident. Furthermore, it remains unclear whether the practical application of these laws results in a frequency of penalty kicks that aligns with the intentions of the governing bodies, or if officials apply stricter or more lenient thresholds in practice. Moreover, handball incidents occur in a variety of contexts, such as after shots on goal, crosses, or during sliding tackles. Each category may involve different justifications for players' body movements, but none are specified in the laws. To promote consistency in the interpretation of handball incidents within and between different competitions in European football, UEFA provides referees

with guidelines on how to interpret various situations [5]. However, the detailed guidelines are not publicly available. Indeed, the challenge of subjective interpretations of game situations is not unique to association football. Research has addressed the relevance and challenges of law interpretations also in Australian football [6,7], rugby union [8] and basketball [9]. Compounding this complexity, a recent meta-analysis across various sports found that decision-making training tends to be less effective for subjective decisions that require higher levels of interpretation (e.g., fouls) than for objective decisions (e.g., offsides, out-of-bounds) [10].

Yet, despite its critical importance and inherent complexity, the interpretation of handball incidents in association football remains an underexplored area within sports science. Existing research has predominantly focused on football referees' interpretation of other law violations, such as fouls [11–14] or offside decisions [15–18]. However, handball decisions represent an independent aspect of the Laws of the Game and must be investigated separately to address the unique controversy surrounding these incidents. To the best of the authors' knowledge, no peer-reviewed academic study has directly addressed the interpretation of handball incidents before. While existing results cannot be directly transferred, work on referees' decision-making regarding other laws can guide research directions for handball incidents. Catteeuw et al. [19] examined foul and offside situations involving 54 professional football referees and found significant differences between referees and assistant referees, with each group demonstrating higher accuracy in their respective decision-making domain. As handball decisions in the penalty area are predominantly the responsibility of the main referee, the influence of role-specific decision-making skills may extend to this area and therefore requires empirical investigation. Moreover, several studies have found significantly higher accuracy in foul decisions among elite referees compared to their sub-elite counterparts [20,21], suggesting that performance level is another factor that should be analysed for handball decisions. Beyond these characteristics of the decision-makers, properties of the decision-making environment, such as video speed have also been shown to influence decision-making of sport officials [22,23]. Research with non-referees found that slow motion inflates the perceived duration of sporting actions [24], while in another study, non-referee participants perceived intent as higher when viewing actions in slow motion [25]. However, Mather and Breivik [26] investigated how video speed affected 80 elite referees' perceptions of intent, finding no significant differences in perceived intent for foul incidents when viewed in slow motion. Given the importance of intent in many handball decisions, the potential influence of slow motion on different participants must be considered when interpreting results.

Furthermore, the perspectives on handball incidents of other key stakeholders, such as coaches and players, remain unknown. As they are directly affected by referees' decisions and frequently voice their opinions in the media, understanding their perspectives is vital to identify the roots of the controversy. Coaches and players possess expert knowledge of player behaviour and specific motor experience [27], which should be beneficial when deciding whether a body movement was natural or not, one factor explicitly named in the law. Some initial research has compared decision-making on tackle incidents across stakeholders. Coleclough [28] found comparable levels of decision accuracy between football coaches and referees. However, the study's small sample size ($n = 13$), with only two participants having professional football experience, limits its generalisability. Similarly, MacMahon et al. [29] found superior decision accuracy in seven elite football referees compared to 34 youth academy players. Gilis et al. [30] investigated interpretations of player-injury incidents among multiple stakeholders in professional football, including 12 referees, 12 players, 12 coaches, and 12 medical staff members, and showed that referees were the most consistent in identifying fouls and assigning sanctions. While these studies offer initial insights, their results cannot be directly applied to handball incidents in professional football. Taken together, they underscore the subjective nature of the Laws of the Game and the importance of comparing how different stakeholders interpret handball incidents. While referees are responsible for enforcing the Laws of the Game impartially, coaches and players often seek to utilise the laws to their advantage [31]. These differing priorities may influence how each group interprets game situations [32]. Therefore, it is important to assess not only whether referees' interpretations align with the laws but also how far their decisions reflect the perspectives of other key stakeholders. Acceptance of a decision does not depend on its accuracy alone [33,34]. Research has highlighted the tension between 'accuracy', which

refers to the correct application of the written law and 'adequacy', which refers to the contextual appropriateness of a decision [33]. In practice, referees utilise game management to balance the technical enforcement of the laws with the flow and spirit of the match to maintain fairness and control [34,35]. Theoretical models suggest that officials use dynamic decision-making thresholds when judging ambiguous incidents [36]. These thresholds are influenced by interacting constraints, such as the match status, player behaviour, or previous decisions, allowing referees to calibrate their judgements to the specific context [37]. It is important to note, however, that the decision-making process in video-based settings differs from the on-field environment. In real matches, decision-making is an emergent behaviour shaped by immediate physical and social constraints, which are often absent or reduced in video tasks [37]. Consequently, video-based assessments may prompt participants to prioritise technical accuracy over game management considerations [37]. Nevertheless, given the inherent subjectivity of the handball law, the concept of adequacy remains crucial even in this controlled setting. Stakeholders likely still evaluate video clips through their internalised thresholds of fairness, judging whether a penalty kick represents a proportionate and therefore adequate outcome [33].

Moreover, consensus among stakeholders plays an important role in decision acceptance. Without a shared understanding, even decisions that are technically accurate may be met with resistance. In the often-heated atmosphere of football matches, off-pitch consensus may not fully eliminate dissent on the field [38]. However, a shared understanding of the Laws is a necessary foundation for acceptance. In team performance literature, this is often referred to as a Shared Mental Model [39]. To gain deeper insight into how such models diverge, it is necessary to investigate the reasons underlying different interpretations of handball incidents [40]. Beyond merely highlighting differences in decision outcomes, identifying the criteria that referees, coaches, and players prioritise during their decision-making process can reveal the origins of these divergent interpretations [41].

In summary, handball incidents in the penalty area are potentially match-deciding events, yet they have received little scientific attention to date. To ensure fair and consistent decision-making, it is essential to investigate how these incidents are interpreted by lawmakers and those who apply the laws. To ensure stakeholder satisfaction, understanding the perspectives of those who are affected by the decisions is vital. Accordingly, the present paper focuses on five constructs central to the interpretation of handball incidents: Accuracy, Strictness, Handball Categories, Consensus, and Reasoning.

## Purpose

Building on the outlined research gap, this article comprises two studies involving key stakeholders in professional association football. Its primary goal is to create a foundational understanding of the controversy surrounding the handball law by investigating the underlying reasons. In doing so, it aims to provide the first empirical evidence to identify potential improvements, enhance educational measures and inform future discussions on revising the law.

**Study 1.** The first study investigates the transition from lawmaking to refereeing practice. While governing bodies establish the Laws of the Game (IFAB) and issue guidelines to referees on their application (UEFA), the extent to which referees successfully translate these theoretical guidelines for handball incidents into consistent decision-making remains an open question. Study 1 assesses the extent to which referees' handball decisions align with UEFA's interpretations (Accuracy) and how frequently they award penalty kicks (Strictness). Given the reported differences in other laws, the study investigates interpretations across refereeing roles as well as across performance levels. Lastly, the study compares different sub-areas of handball incidents (Handball Categories) to identify key areas of action.

**Study 2.** The second study expands the scope to the inner circle of football practitioners consisting of referees, coaches, and players. It compares Accuracy and Strictness across these stakeholders, assesses agreement of interpretations within and across those groups (Consensus), and analyses the reasons behind stakeholders' decision-making (Reasoning) to gain deeper insights into the factors shaping their interpretations. The objective is to identify systematic divergences that undermine the acceptance of handball decisions in practice and derive actionable implications to reduce the controversy.

## Methods

### Setting

The two survey-based studies were conducted online via CoreXM (Qualtrics LLC, Utah, USA). Participants were shown video scenes of penalty area handball incidents and asked to provide their responses. Survey 1, targeting referees, was conducted before the start of the 2023/24 season between 1 and 19 June 2023, whereas Survey 2, involving coaches and players, took place during the 2023/24 season between 29 February and 31 May 2024. In both surveys, participants were instructed to consider *The Laws of the Game 23/24* [2] as the most recent version of the laws at the time. As the handball law did not change in the period between the two surveys, the results are expected to be comparable, and referees' replies to Survey 1 were also used in the analysis of Study 2. Participants remained anonymous, were informed about the procedures and data protection measures in place, and provided written informed consent. The study received written ethical approval (A2.5.4-307_vb) from the Ethics Committee of the Faculty of Economics and Social Sciences at the University of Tübingen.

### Sample

A total of $N = 231$ participants ($M_{age} = 37.9 \pm 12.0$ years; 5 female) were part of the research. In Study 1, all referees involved in German men's professional football at the time of the study were invited to participate, with $n = 154$ ($M_{age} = 39.0 \pm 11.9$ years; 5 female) completing Survey 1. German men's professional football comprises the top three national divisions: Bundesliga (1st division), 2. Bundesliga (2nd division), and 3. Liga (3rd division) [42]. Consistent with recent recommendations for standardised reporting in officiating research, these divisions correspond to the '(1) international/professional' competition level suggested by Webb et al. [43]. Referees were classified into three *Referee Role Groups*: REF ($n = 46$; $M_{age} = 33.6 \pm 7.5$ years; 2 female), including main referees who officiate in the centre of the field; AREF ($n = 74$; $M_{age} = 34.2 \pm 7.2$ years; 3 female), comprising assistant referees and VARs who support the main referee; and REFI ($n = 34$; $M_{age} = 56.6 \pm 7.2$ years; all male), consisting of referee instructors – committee members, referee coaches, and observers – responsible for education and evaluation. The second group classification by *Referee Performance Level* was based on the highest league in which a referee was active at the time of data collection: 1st division ($n = 50$; $M_{age} = 39.5 \pm 5.4$ years; all male), 2nd division ($n = 28$; $M_{age} = 34.1 \pm 4.2$ years; 1 female), or 3rd division ($n = 42$; $M_{age} = 27.4 \pm 5.0$ years; 4 female). The REFI group was excluded from this analysis as they no longer officiate professionally. Detailed demographic information, including participant frequency within standardised age categories and experience categories [43], is provided in the supporting information (S4 Table).

In Study 2, all coaches employed in German men's professional football, as well as German coaches working in European men's professional football within the two years preceding data collection, were invited to the Bundesliga Coaches Congress, organised by the Association of German Football Coaches (Bund Deutscher Fußball-Lehrer e.V.). They were contacted in advance to participate in the study, with $n = 31$ professional football coaches ($M_{age} = 48.0 \pm 9.0$ years; all male) completing Survey 2. Additionally, all clubs participating in German men's professional football during the 2023/24 season were invited to involve their first men's teams, with $n = 46$ professional football players ($M_{age} = 27.3 \pm 4.2$ years; all male) completing Survey 2. Referees, coaches, and players were analysed as whole *Stakeholder Groups*, without further classification by roles or performance levels.

### Measures and procedures

Both studies used video scenes of handball incidents from professional international football matches, ensuring that no participant had personal involvement. Clips were obtained from television broadcasts and included the match clock, score and stadium noise to preserve realism, but commentary was excluded to avoid bias from the commentator's opinion. Each video showed one handball incident involving a defender's hand or arm touching the ball inside their team's penalty area,

potentially leading to a penalty kick for the attacking team. Replays and slow-motion views from multiple angles were provided to present participants with all necessary information for their decisions. Replays started immediately after the incident, thereby minimising the potential visibility of original match referees' decisions. The interpretations of all handball incidents based on the Laws of the Game by the UEFA Referees Committee were available to the research team. Half of the selected scenes contained punishable and the other half not punishable handball incidents according to UEFA's interpretations. Participants were unaware of UEFA's interpretations or the equal distribution. Some scenes also depicted an on-field review (OFR) due to VAR interventions in the original match ($m = 6$ in Study 1; $m = 3$ in Study 2), but the video scenes were cut before the match referees' final decisions were visible. To assess potential confounding effects, dependent-samples $t$-tests were conducted to compare scenes with and without OFRs. No significant differences were found in participants' Accuracy (Study 1: $t(153) = 1.57$, $p = .12$; Study 2: $t(230) = 0.88$, $p = .38$), so this factor was not further considered.

Participants viewed the video scenes in individually randomised orders and could replay them as often as desired. They were asked to decide whether each incident was punishable or not punishable according to the Laws of the Game (Q1), provide reasoning for their decision (Q2), and rate the difficulty of the decision on a scale from 1 (very easy) to 10 (very difficult) (Q3). In Study 1, referees viewed $m = 30$ video scenes ($M_{length} = 22.5 \pm 6.3\,s$) and completed the survey in approximately 30 minutes. They provided their reasoning (Q2) in a free-text box. Mean difficulty ratings were calculated for the 30 scenes, with an overall mean difficulty rating of $4.2 \pm 2.7$. In Study 2, a reduced set of $m = 18$ scenes ($M_{length} = 22.9 \pm 6.0\,s$) was used to increase response rates among coaches and players while maintaining comparability to Study 1, shortening the survey duration to approximately 15 minutes. In Study 2, participants from all three Stakeholder Groups rated the difficulty of the remaining 18 handball incidents with a mean of $4.1 \pm 2.6$, which was comparable to Study 1. To simplify the reasoning response (Q2), participants could select from predefined options or enter their reasoning in a text box. Additionally, recognising that coaches and players might have less theoretical knowledge of the Laws of the Game, the text of the handball law was provided and could be displayed above each video if desired.

## Handball Categories

Based on discussions with experts from the German Football Association (DFB) referee department, the video scenes were categorised to describe recurring patterns of handball incidents in the penalty area, frequently observed in professional football. The same six categories were applied in both Study 1 and Study 2, with differences in the number of scenes per category due to the reduced number of scenes in Study 2:

- *Arm supporting body*: A defender's hand or arm touches the ball while being used to support their body during a fall or sliding tackle ($m = 3$ in Study 1; $m = 2$ in Study 2).

- *Blocked cross*: A defender's hand or arm blocks a cross ($m = 4$ in Study 1; $m = 3$ in Study 2).

- *Blocked shot*: A defender's hand or arm blocks a shot directed towards the goal ($m = 7$ in Study 1; $m = 4$ in Study 2).

- *Deflection*: The ball touches a player's hand or arm after deflecting off their own body or that of a teammate immediately before the handball incident ($m = 6$ in Study 1; $m = 5$ in Study 2).

- *Header from behind*: A defender unsuccessfully attempts to head the ball out of the penalty area, allowing an attacker behind him to head the ball towards the goal, which is then blocked by the defender's outstretched hand or arm ($m = 3$ in Study 1; $m = 2$ in Study 2).

- *Other*: Incidents not fitting into the above categories but still involving a defender's hand or arm touching the ball in the penalty area ($m = 7$ in Study 1; $m = 2$ in Study 2).

### Reasoning Categories.

From referees' free-text responses in Survey 1, 17 recurring reasons for handball decisions were identified (Table 1). Each response was assigned manually by the first author to one of these reasons, with the first-mentioned reason considered primary if multiple reasons were given. These 17 reasons were presented as single-choice options in Survey 2 for coaches and players, who selected their primary reason. A free-text option was also available for responses not aligning with the predefined choices. For further analysis, the 17 recurring reasons were grouped into four overarching Reasoning Categories: *Intent*, *Naturalness*, *Avoidability*, and *Impact*. The first two categories relate directly to the Laws of the Game, while the latter two address situational context and are not explicitly mentioned in the Laws. These categories comprise reasons for both punishable and not punishable decisions. Free-text responses that deviated from the 17 recurring reasons were manually assigned to one of the four Reasoning Categories, where possible. Unclassifiable responses were excluded (1.17% of total Q2 data), resulting in $k = 4{,}117$ responses included in the analysis.

### Outcome variables

Five outcome variables were derived from the three survey questions that participants answered for each handball incident. Each variable addressed a different aspect of handball interpretation.

**Q1: "Was this handball punishable or not punishable according to the Laws of the Game?".**

- *Accuracy* referred to the percentage of decisions that aligned with UEFA's interpretation. For each participant, the number of accurate responses was divided by the total number of scenes. For example, a referee who agreed with UEFA in 27 out of 30 scenes achieved 90% Accuracy. This variable was analysed in both studies.

- *Strictness* indicated how frequently a participant classified incidents as punishable. It reflected the participant's overall tendency to award penalty kicks, independent of Accuracy. For instance, a coach who deemed 7 out of 18 handball incidents punishable reached 38.9% Strictness. By design, UEFA's Strictness was 50% and served as a baseline. This variable was analysed in both studies.

- *Consensus* captured the level of agreement within Stakeholder Groups for each scene. It was determined by the distribution of punishable vs. not punishable decisions per scene within each group. For example, if 132 referees classified a scene as punishable and 22 as not punishable, the within-group Consensus among referees was 85.7%/14.3%. This variable was analysed only in Study 2.

**Table 1. Recurring reasons and overarching Reasoning Categories.**

| Reasoning Category | Punishable | Not punishable |
|---|---|---|
| Intent | Intention to stop the ball/ not trying to avoid handball | No intention to stop the ball/ trying to avoid handball |
| | Active movement of hand or arm towards the ball or ball's trajectory | Accidentally hitting own hand or arm when shooting or heading the ball |
| Naturalness | Unnatural hand or arm position during this body movement makes body bigger | Natural hand or arm position during this body movement |
| | | Body is not made bigger/ hand or arm is close to the body |
| | Hand or arm is tense | Hand or arm is not tense |
| Avoidability | Player has orientation to the ball/ knows where the ball is | Player has no orientation to the ball/ does not know where the ball is |
| | Ball has been travelling for a long time | Ball coming from short distance |
| | Predictable trajectory of the ball | Unexpected trajectory of the ball |
| Impact | A shot on goal or goal-scoring opportunity was prevented | There was no goal-scoring opportunity |

**Q2: "What was the primary reason for your decision?".**

- *Reasoning Distribution* represented the proportion of reasons selected by each Stakeholder Group across the four Reasoning Categories: Intent, Naturalness, Avoidability, and Impact. For example, players selected 225, 298, 184, and 113 reasons from each category, respectively, resulting in a Reasoning Distribution of 27.4%, 36.3%, 22.4%, and 13.8%. This variable was analysed only in Study 2.

**Q3: "How difficult was this handball decision, from 1 (very easy) to 10 (very difficult)?".**

- *Difficulty* was defined as the mean difficulty rating per video scene, based on all participant responses in Study 1. This variable was used to compare Handball Categories among referees and to guide scene selection for Study 2, ensuring that the overall mean difficulty rating of the remaining 18 scenes closely matched the overall mean difficulty rating of the full set of 30 scenes.

## Statistical analysis

Data were analysed using Python v3.12 in PyCharm 2024.2 Professional Edition (JetBrains s.r.o., Prague, Czech Republic). The following Python libraries were used: pandas v2.2.3, NumPy v1.26.4, SciPy v1.14.1, statsmodels v0.14.4, Pingouin v0.5.5, and Matplotlib v3.9.2. The significance level was set at $\alpha = .05$ and two-tailed testing was applied. Hedges' $g$, $\eta^2$, and Cramér's $V$ were calculated as measures of effect size and classified according to Cohen [44] as small ($0.20 \leq g \leq 0.49$; $0.01 \leq \eta^2 \leq 0.05$; $0.10 \leq V \leq 0.29$), medium ($0.50 \leq g \leq 0.79$; $0.06 \leq \eta^2 \leq 0.13$; $0.30 \leq V \leq 0.49$) or large ($g \geq 0.80$; $\eta^2 \geq 0.14$; $V \geq 0.50$). Post hoc sensitivity analyses ($\alpha = .05$, $1 - \beta = .80$) were conducted using G*Power v3.1.9.7 [45]. The analyses indicated sensitivity to detect medium effects for comparisons between Referee Performance Levels ($n = 120$) and Referee Role Groups ($n = 154$), and small effects for Stakeholder Groups ($N = 231$).

Descriptive statistics including means, standard deviations, medians, and ranges were computed across groups. Levene's test and the Shapiro-Wilk test assessed homogeneity of variances and normality, respectively. If assumptions were violated for sub-datasets, non-parametric alternatives were conducted as a cross-check: the Kruskal-Wallis test for one-way analysis of variance (ANOVA), Dunn's test for Tukey's honestly significant difference (HSD) test, the Friedman test for repeated-measures ANOVA, the Wilcoxon signed-rank test for the dependent $t$-test, and Fisher's exact test for chi-square tests. Except for one post hoc analysis, parametric and non-parametric tests yielded identical significance levels. Therefore, parametric results are reported throughout the article, but one post hoc analysis reports non-parametric results of the Wilcoxon signed-rank test instead.

Differences in Accuracy and Strictness across Referee Role Groups, Referee Performance Levels, and Stakeholder Groups were assessed using one-way ANOVAs. Significant results were followed by post hoc Tukey's HSD tests. To assess differences in Strictness between groups of participants and UEFA's baseline of 50%, one-sample $t$-tests were conducted. For subgroup comparisons of the three Referee Role Groups and three Referee Performance Levels with UEFA, Bonferroni-adjusted $p$-values were calculated to control for multiple comparisons. In Study 1, mean Accuracy and mean Difficulty per participant were additionally calculated for each Handball Category and differences across Handball Categories were analysed using Repeated Measures ANOVAs to account for within-subject comparisons. Wilcoxon signed-rank tests for Accuracy (as this was the only analysis where parametric and non-parametric results differed) and dependent $t$-tests for Difficulty (each Bonferroni-adjusted) were used for post hoc analyses. Strictness was not assessed further, as UEFA's baseline of 50% was not applicable to individual Handball Categories.

Since Study 2 focused on the agreement between Stakeholder Groups, Consensus for single handball incidents was analysed across them. A contingency table was created for each video scene, counting the decisions punishable and not punishable per Stakeholder Group. A chi-square test was performed for each scene, followed by pairwise chi-square tests between Stakeholder Groups as post hoc analyses with Bonferroni adjustments applied.

For the analysis of Reasoning Distribution, the total number of replies for each Reasoning Category (Intent, Naturalness, Avoidability, and Impact) was summed up per Stakeholder Group in a contingency table. A chi-square test was performed to examine differences in Reasoning Distribution between the Stakeholder Groups. Adjusted standardised residuals (ASRs) were calculated for post hoc analysis to identify which Reasoning Categories significantly differed from the expected frequencies per Stakeholder Group. The ASRs were computed using the following formula:

$$ASR = \frac{Observed\ Frequency\ - Expected\ Frequency}{\sqrt{Expected\ Frequency * \left(1 - \frac{Row\ Total}{Grand\ Total}\right) * \left(1 - \frac{Column\ Total}{Grand\ Total}\right)}}$$

Bonferroni adjustment was applied to account for a total of 12 comparisons. In accordance with the two-tailed testing applied, ASRs with absolute values greater than 2.87 were considered statistically significant. Odds ratios (*OR*) were calculated as effect size measures within Reasoning Categories between the Stakeholder Groups.

## Results

### Study 1

**Accuracy.** Participants' mean Accuracy score for the 30 handball incidents in Survey 1 was 84.0±7.3%, with a median of 83.3%, and individual scores ranging from 60.0% to 100.0%. While two participants reached full accordance with UEFA (100% Accuracy), all subgroups remained below this score (Table 2). Significant differences were found across Referee Role Groups ($F(2, 151) = 3.24$, $p = .04$, $\eta^2 = .04$) and across Referee Performance Levels ($F(2, 117) = 3.29$, $p = .04$, $\eta^2 = .05$), both with small overall effect sizes. Post hoc analyses revealed significantly higher Accuracy for REF than for AREF ($p = .04$, $g = 0.48$), and for 2nd than for 3rd division officials ($p = .03$, $g = 0.62$), with small and medium effect sizes, respectively. No significant differences were found between REF and REFI ($p = .18$), AREF and REFI ($p = .95$), 1st- and 2nd-division officials ($p = .42$), or 1st- and 3rd-division officials ($p = .28$).

**Table 2. Accuracy and Strictness in Study 1.**

| Referee Role Group | n | Accuracy M±SD [%] | Hedges' g vs. REF | vs. AREF | Strictness M±SD [%] | Hedges' g vs. UEFA | vs. REF | vs. AREF |
|---|---|---|---|---|---|---|---|---|
| Total | 154 | 84.0±7.3 | | | 42.9±9.5 | −0.75*** | | |
| REF | 46 | 86.2±7.2 | – | | 43.0±8.8 | −0.79*** | – | |
| AREF | 74 | 82.9±6.9 | −0.48* | – | 41.5±10.0 | −0.85*** | −0.16 | – |
| REFI | 34 | 83.3±8.0 | −0.38 | 0.06 | 45.9±8.8 | −0.47** | 0.32 | 0.45 |
| **Referee Performance Level** | | | vs. 1st | vs. 2nd | | vs. UEFA | vs. 1st | vs. 2nd |
| Total | 120 | 84.2±7.1 | | | 42.1±9.5 | −0.83*** | | |
| 1st division | 50 | 84.5±7.1 | – | | 40.3±10.2 | −0.95*** | – | |
| 2nd division | 28 | 86.6±6.6 | 0.30 | – | 43.0±7.2 | −0.97*** | 0.28 | – |
| 3rd division | 42 | 82.2±7.1 | −0.31 | −0.62* | 43.7±9.9 | −0.64*** | 0.33 | 0.07 |

Note. Descriptive statistics and effect sizes for Accuracy and Strictness in Study 1, separated by Referee Role Group and Referee Performance Level. Comparisons with UEFA based on one-sample t-tests; subgroup differences based on Tukey's HSD.

* p<.05, ** p<.01, *** p<.001.

**Strictness.** Participants' mean Strictness was 42.9 ± 9.5%, with a median of 43.3% and individual scores ranging from 23.3% to 70.0%. The overall Strictness was significantly lower than UEFA's baseline (50%), with a medium effect size ($t$(153) = −9.25, $p < .001$, $g = −0.75$). Post hoc analyses confirmed lower Strictness than UEFA's baseline for all subgroups, with small to large effect sizes (each $p < .01$, $g = 0.47–0.97$). No significant differences in Strictness were found across Referee Role Groups ($p = .08$) or across Referee Performance Levels ($p = .22$).

### Handball Categories.

Participants reached significantly different Accuracy scores across the six Handball Categories, with a large overall effect size ($F$(5, 765) = 27.31, $p < .001$, $η² = .15$). The highest Accuracy was observed in the category Arm supporting body (92.9 ± 14.7%), followed by Blocked shot (88.1 ± 11.8%), Other (86.2 ± 14.1%), Deflection (82.0 ± 13.1%), Header from behind (79.2 ± 26.2%) and the lowest Accuracy in Blocked cross (72.7 ± 19.9%). The post hoc analysis revealed significant differences in a total of eleven pairwise comparisons (each $p < .04$, Bonferroni-adjusted). Effect sizes ranged from small to large ($g = 0.35–1.15$), with the largest effect size observed between Arm supporting body and Blocked cross (Fig 1).

Difficulty ratings also varied significantly, with a medium overall effect size ($F$(5, 765) = 20.39, $p < .001$, $η² = .12$). Highest Difficulty was observed in Blocked cross (4.7 ± 1.6), followed by Header from behind (4.4 ± 1.8), Other (4.4 ± 1.3), Deflection (4.1 ± 1.5), Blocked shot (3.8 ± 1.3) and lowest Difficulty in Arm supporting body (3.6 ± 1.8). Post hoc analysis revealed significant differences in a total of ten pairwise comparisons (each $p < .03$, Bonferroni-adjusted). Effect sizes ranged from small to medium ($g = 0.22–0.67$), with the largest again between Arm supporting body and Blocked cross.

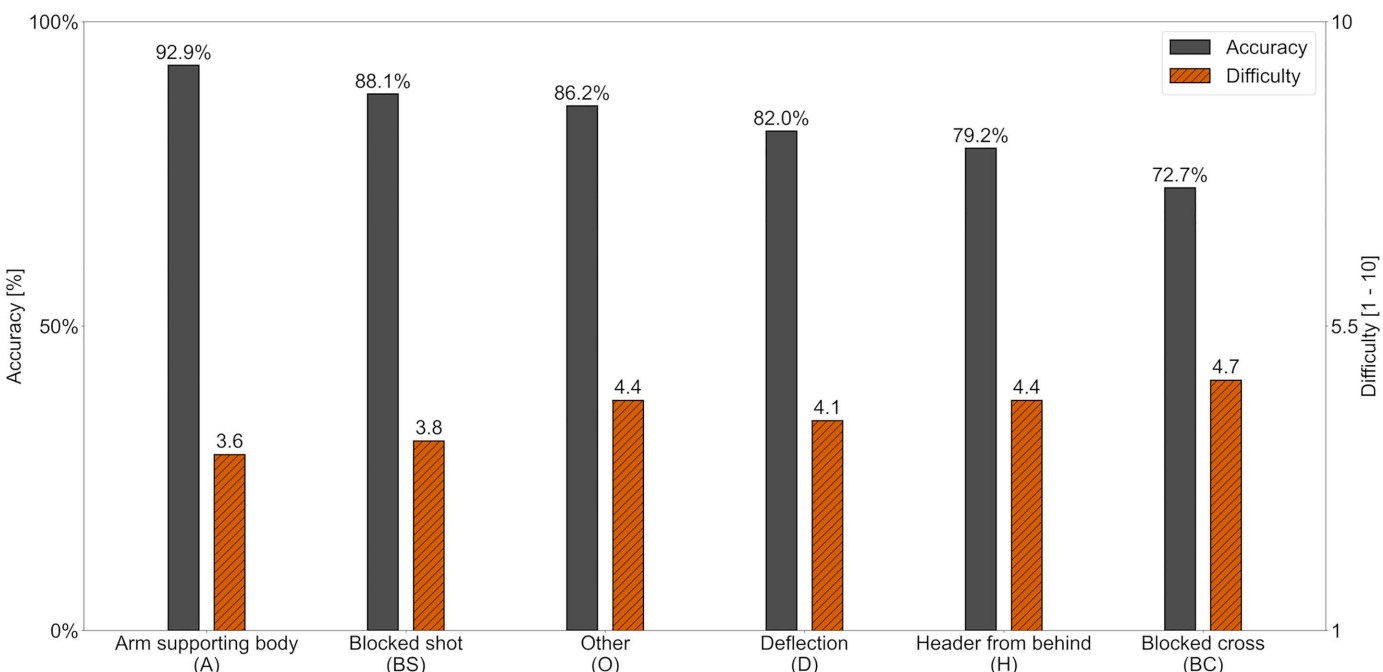

**Fig 1. Accuracy and Difficulty of Handball Categories in Study 1.** Note. Mean Accuracy and mean Difficulty of participants per Handball Category in Study1. Categories are sorted by mean Accuracy in descending order. Pairwise significant differences ($p < .05$, Bonferroni-adjusted): Accuracy: A > O, D, H, BC; BS > D, H, BC; O > D, H, BC; D > BC. Difficulty: BC > O, D, BS, A; H > BS, A; O > BS, A; D > BS, A.

## Study 2

**Accuracy.** Significant differences in Accuracy were found between referees, coaches, and players, with a large overall effect size ($F_{(2, 228)} = 67.65$, $p < .001$, $\eta^2 = .37$). The post hoc analysis revealed significantly higher Accuracy for referees than for coaches ($p < .001$, $g = 1.88$) and players ($p < .001$, $g = 1.46$), both with large effect sizes. No significant differences were found between coaches and players ($p = .22$) (Table 3).

**Strictness.** Significant differences were also found in Strictness between the Stakeholder Groups, with a small overall effect size ($F_{(2, 228)} = 6.25$, $p = .002$, $\eta^2 = .05$). The post hoc analysis revealed significantly higher Strictness for referees than for players, with a medium effect size ($p = .004$, $g = 0.56$). No significant differences were found between referees and coaches ($p = .12$) or coaches and players ($p = .76$).

**Consensus.** Within-group Consensus for single handball decisions varied markedly, with some scenes even reaching extreme values. Maximum Consensus was observed by coaches in scenes 2 and 19 (both Deflection), with a 0%/100% decision distribution. In contrast, minimum within-group Consensus was observed by referees in scene 4 (Blocked cross) and by players in scene 9 (Header from behind), both with a 50%/50% decision distribution. Other scenes ranged between those extreme distributions.

The comparison between Stakeholder Groups revealed significantly different decision distributions in 11 out of the 18 handball incidents, with small to medium overall effect sizes (each $p < .019$, $V = .19–.42$) (Table 4). Post hoc analyses revealed significant differences between referees and coaches in nine handball incidents (each $p < .036$, Bonferroni-adjusted; $V = .18 –.44$, small to medium effects), between referees and players in eight handball incidents (each $p < .042$, Bonferroni-adjusted; $V = .17–.42$, small to medium effects), and between coaches and players in one handball incident ($p = .012$, Bonferroni-adjusted; $V = .33$, medium effect). Significant differences between Stakeholder Groups were observed across all Handball Categories, while the frequency of handball incidents with differences in Consensus varied:

- Arm supporting body: 2 out of 2 incidents

- Blocked cross: 1 out of 3 incidents

- Blocked shot: 4 out of 4 incidents

- Deflection: 1 out of 5 incidents

- Header from behind: 1 out of 2 incidents

- Other: 2 out of 2 incidents

**Reasoning distribution.** Significant differences were found in the overall Reasoning Distribution between the Stakeholder Groups, with a small overall effect size ($\chi^2_{(6)}$, $k = 4117$) $= 426.66$, $p < .001$, $V = .23$). The post hoc analysis

**Table 3. Accuracy and Strictness in Study 2.**

| Stakeholder Group | $n$ | Accuracy | | | Strictness | | |
|---|---|---|---|---|---|---|---|
| | | $M \pm SD$ [%] | Hedges' $g$ | | $M \pm SD$ [%] | Hedges' $g$ | |
| | | | vs. Referees | vs. Coaches | | vs. Referees | vs. Coaches |
| Total | 231 | 76.3 ± 12.1 | | | 38.9 ± 13.1 | | |
| Referees | 154 | 81.5 ± 9.2 | – | | 41.0 ± 11.6 | – | |
| Coaches | 31 | 63.8 ± 10.2 | −1.88*** | – | 36.0 ± 14.7 | −0.41 | – |
| Players | 46 | 67.5 ± 10.6 | −1.46*** | 0.35 | 33.9 ± 15.2 | −0.56** | −0.14 |

Note. Descriptive statistics and effect sizes of Accuracy and Strictness in Study 2, by Stakeholder Group.

\* p < .05, \*\* p < .01, \*\*\* p < .001.

**Table 4. Frequencies of punishable decisions and chi-square test statistics in Study 2.**

| Handball Category | Scene no. | Punishable decisions Referees (n=154) n | % | Coaches (n=31) n | % | Players (n=46) n | % | Omnibus test χ2 | p | Cramér's V | Referees vs. Coaches Cramér's V | Referees vs. Players | Coaches vs. Players |
|---|---|---|---|---|---|---|---|---|---|---|---|---|---|
| Arm supporting body | 16 | 6 | 3.9 | 6 | 19.4 | 8 | 17.4 | 13.34 | .0013 | .24** | .21* | .20* | .00 |
| | 27 | 130 | 84.4 | 17 | 54.8 | 21 | 45.7 | 32.61 | <.001 | .38*** | .26** | .37*** | .06 |
| Blocked cross | 4 | 77 | 50.0 | 10 | 32.3 | 19 | 41.3 | 3.76 | .15 | .13 | .12 | .06 | .06 |
| | 6 | 66 | 42.9 | 15 | 48.4 | 22 | 47.8 | 0.56 | .75 | .05 | .03 | .03 | .00 |
| | 22 | 132 | 85.7 | 17 | 54.8 | 19 | 41.3 | 41.00 | <.001 | .42*** | .27*** | .42*** | .11 |
| Blocked shot | 1 | 1 | 0.7 | 4 | 12.9 | 6 | 13.0 | 17.23 | <.001 | .27*** | .24** | .25** | .00 |
| | 7 | 143 | 92.9 | 28 | 90.3 | 34 | 73.9 | 12.82 | .0016 | .24** | .01 | .23** | .17 |
| | 11 | 121 | 78.6 | 16 | 51.6 | 32 | 69.6 | 9.93 | .0070 | .21** | .21* | .08 | .15 |
| | 29 | 6 | 3.9 | 13 | 41.9 | 5 | 10.9 | 40.12 | <.001 | .42*** | .44*** | .10 | .33* |
| Deflection | 2 | 2 | 1.3 | 0 | 0.0 | 2 | 4.4 | 2.57 | .28 | .11 | .00 | .05 | .05 |
| | 19 | 1 | 0.7 | 0 | 0.0 | 1 | 2.2 | 1.27 | .53 | .07 | .00 | .00 | .00 |
| | 24 | 56 | 36.4 | 9 | 29.0 | 12 | 26.1 | 1.98 | .37 | .09 | .04 | .08 | .00 |
| | 26 | 128 | 83.1 | 19 | 61.3 | 27 | 58.7 | 15.16 | <.001 | .26*** | .18* | .23** | .00 |
| | 28 | 19 | 12.3 | 4 | 12.9 | 3 | 6.5 | 1.30 | .52 | .08 | .00 | .06 | .06 |
| Header from behind | 9 | 95 | 61.7 | 13 | 41.9 | 23 | 50.0 | 5.15 | .076 | .15 | .13 | .09 | .05 |
| | 10 | 122 | 79.2 | 10 | 32.3 | 26 | 56.5 | 30.08 | <.001 | .36*** | .37*** | .20* | .21 |
| Other | 14 | 23 | 14.9 | 14 | 45.2 | 15 | 32.6 | 16.87 | <.001 | .27*** | .26*** | .17* | .10 |
| | 20 | 8 | 5.2 | 6 | 19.4 | 6 | 13.0 | 7.94 | .019 | .19* | .17 | .11 | .05 |

Note. Frequencies of punishable decisions and chi-square test statistics in Study 2, by scene number, ordered by Handball Category. For clarity, only punishable decisions are reported, as not punishable decisions can be easily inferred.

\* p<.05, \*\* p<.01, \*\*\* p<.001, with Bonferroni-adjusted post hoc comparisons.

revealed significant differences in three of the four Reasoning Categories (Fig 2). Reasons categorised in Naturalness were significantly more frequently named by referees and less frequently by coaches and players than expected (each $|ASR| > 7.42$), with the largest differences between referees and coaches (OR = 2.44). Reasons categorised in Avoidability were significantly less frequently named by referees and more frequently by coaches and players than expected (each $|ASR| > 4.70$), with the largest differences between coaches and referees (OR = 1.95). Reasons categorised in Impact were significantly less frequently named by referees and more frequently by coaches and players than expected (each $|ASR| > 4.62$), with the largest differences between players and referees (OR = 33.53). No significant differences were observed in the Reasoning Category Intent (each $|ASR| < 2.28$).

## Discussion

Using a video-based design, the present research is the first to provide systematic insights into the controversy surrounding the interpretation of handball incidents in professional association football. With n=154 referees from a leading European football nation, Study 1 is the largest of its kind to date (for a comparison of sample sizes, see reviews [10,46,47]). It examined how effectively referees implement the guidelines provided by governing bodies in practice. Study 2 investigated how closely referees' interpretations align with those of other key participants in the game – coaches and players – each with professional experience at the highest levels of competition. Given the decisive nature of handball decisions in the penalty area, this research offers a unique opportunity to identify inconsistencies in interpretation and to highlight potential areas for improvement in the application of the handball law.

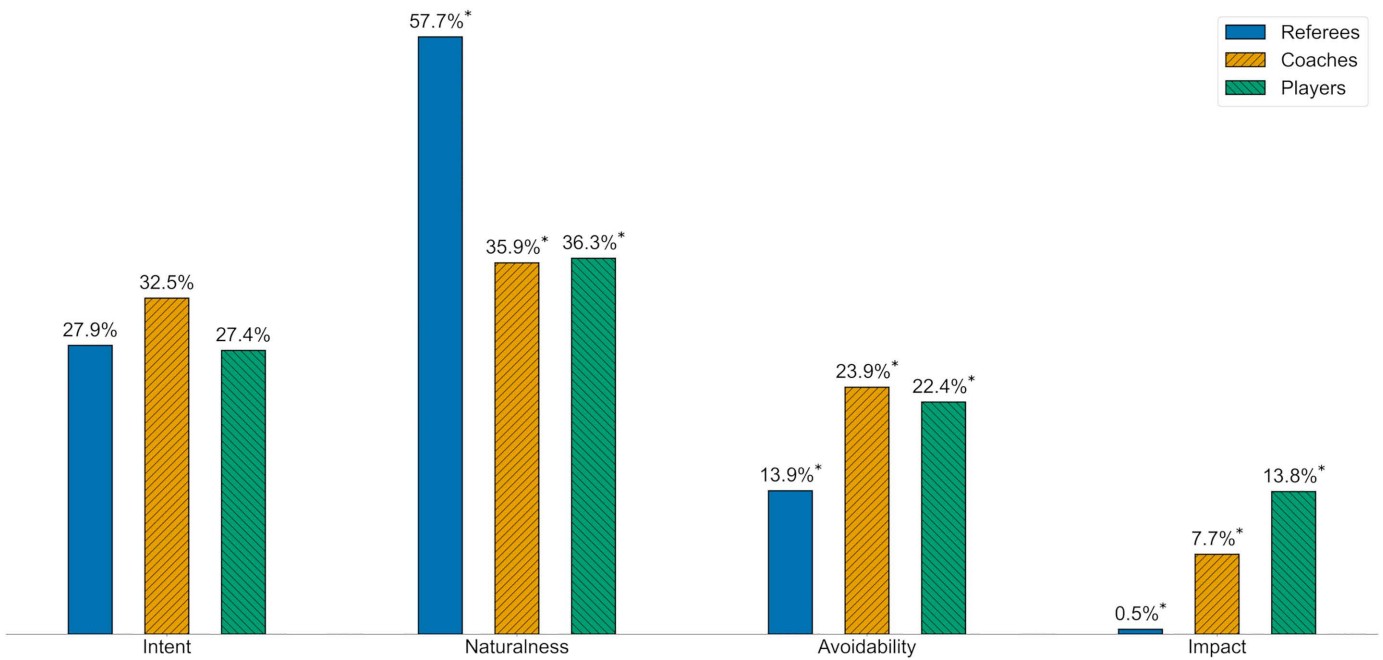

**Fig 2. Reasoning Distribution in Study 2.** Note. Mean Reasoning Distribution separated by Stakeholder Groups.

## Study 1

Although the total mean Accuracy of 84.0% indicates that one in six handball decisions by referees in this study was not in accordance with UEFA's guidelines, this represents a higher level of Accuracy than that reported in other video-based studies on foul decisions [41]. Main referees' handball decisions were slightly more aligned with UEFA's interpretations compared to those of assistant referees and VARs, consistent with previous research on role-specific decision-making in officiating [19]. Given that handball decisions in the penalty area are predominantly the responsibility of the main referee, their slightly higher Accuracy may be expected. While the effect size remained small, any discrepancy could present challenges in joint decision-making during matches, particularly in situations where assistant referees or VARs provide input that differs from the main referee's interpretation [48]. Research emphasises that effective officiating team performance relies on Shared Mental Models [49] and referee education should therefore aim to align handball decision-making across referee roles as much as possible to ensure a coherent team performance [39].

In addition to role-based differences, performance level was also associated with the observed Accuracy. A medium-sized difference was found between 2nd- and 3rd-division officials. Although 2nd-division officials also had a slightly higher mean Accuracy than 1st-division officials, this difference was not statistically significant. Referees at higher levels of professional football typically receive more extensive training, including greater exposure to UEFA's guidelines. As decision-making is a fundamental skill of sport officials [43] and required for promotion to higher leagues, the lower Accuracy observed in the 3rd division appears reasonable and is consistent with previous research [20,21]. It could be hypothesised that the presence of the VAR in the German 1st and 2nd divisions, while it is unavailable in the 3rd division, contributes to these results, as these officials might be more accustomed to video-based decision-making. However, we do not consider the availability of the VAR system during matches to be a relevant determinant of the observed performance differences in this study. In German professional football, referees in all three divisions perform extensive video analysis on a weekly basis. Furthermore, performance evaluation by referee observers across all three divisions relies

heavily on video footage. Therefore, high familiarity with video decision-making can be assumed for all groups. The difference in Accuracy is more likely a reflection of a generally higher decision quality, which is a main criterion for promotion, rather than a result of the VAR technology. Consequently, regarding decision-making training, we recommend focusing on complex situations characterised by high ambiguity and difficulty. Learning to identify the relevant cues is needed for accurate decision-making [50]. Aligning the decision-making thresholds of lower-level referees with those of the highest level in these specific situations is essential to prepare them for potential promotion [51].

On average, referees applied a lower level of Strictness compared to UEFA's baseline. Given the high Accuracy, this lower Strictness indicates that referees tended to interpret incidents more leniently, deciding against a penalty kick in ambiguous situations. In combination with results of Study 2, this suggests that referees take an intermediate position between governing bodies and coaches and players in terms of how strictly they apply the handball law. This tendency suggests that even in a video-based setting, referees might subconsciously prioritise perceived decision adequacy over strict technical accuracy [33]. By calibrating their decision-making thresholds [36], they may attempt to balance rule enforcement with game management [52]. The tendency to avoid decisions that significantly influence the match outcome is a known phenomenon referred to as 'impact aversion', which has also been observed in other sports such as Australian football [53]. This raises the question whether UEFA's guidelines reflect the expectations towards officiating in practice. It may be worth reconsidering these guidelines and the frequency of penalty kicks that should be awarded for handball incidents. In this regard, it is essential to take the specific characteristics of each Handball Category into account.

Across the six Handball Categories, significant differences were observed. Categories with higher Accuracy tended to receive lower Difficulty ratings, and vice versa, suggesting that referees are sensitive to the underlying complexity of different handball decisions. Arm supporting body achieved the highest Accuracy, likely due to well-defined criteria that are straightforward to apply in situations with few diagnostic cues in a video-based setting, resulting in the lowest Difficulty. For instance, when a defender's arm is positioned vertically towards the ground to support the body, it is explicitly considered natural and therefore not punishable under current UEFA guidelines. In contrast, Blocked cross proved the most challenging category, combining the lowest Accuracy with the highest Difficulty. These situations tend to be more complex, involving multiple ambiguous cues, such as outstretched arms used for balance while running or jumping, variation in ball–defender distance, or arm tension. The complexity of these situations presumably places higher demands on referees' visual search behaviours [20] and attention allocation [54] to identify the relevant diagnostic cues. In addition to improving Accuracy among referees, clearer guidance would also provide players with greater certainty on how to position their arms without risking a penalty kick against their team, as many defenders currently hide their hands behind their backs [55]. One approach would be to introduce actionable criteria, comparable to those used for Arm supporting body. These should distinguish between punishable and non-punishable arm positions when blocking crosses, with explicit definitions of allowed and disallowed positions.

## Study 2

Study 2 demonstrated that referees make handball decisions significantly more in accordance with UEFA's interpretations than coaches and players. This finding contrasts with Coleclough [28], who observed similar decision accuracy between coaches and referees for tackle incidents, but aligns with MacMahon et al. [29], who found that referees outperformed players in decision accuracy. However, the present study is not fully comparable to MacMahon et al. [29], as their sample consisted of youth players rather than professionals and also considered tackle incidents. The greater alignment of referees with UEFA's guidelines is expected, as referees receive ongoing training based on these guidelines, whereas coaches and players are not systematically trained in the same way. Moreover, referees not only learn theoretically about the guidelines but also frequently train decision-making with respect to the Laws of the Game in video-based settings [56]. Schweizer et al. [50] suggested that taking the relevant cues – and only these – into account could be improved by video-based training, thereby increasing decision accuracy. Additionally, the location of referees' visual attention and their

gaze behaviour can produce substantial differences in what they perceive and thus in the information available for their decisions [54,57]. In contrast, coaches and players do not regularly train decision-making with respect to the Laws of the Game [58]. Their interpretations may be influenced more by perceptions of fairness [59], and the specific match context rather than strict adherence to the Laws of the Game or guidelines by governing bodies [60].

Players exhibited the lowest Strictness among the three Stakeholder Groups and would have awarded significantly fewer penalty kicks for handball incidents than referees. This discrepancy likely contributes to the ongoing controversy surrounding the handball law. Given the high conversion rate of penalty kicks and their potential to influence match outcomes, it is understandable that differences in Strictness lead to emotional reactions. On the field, the interest in gaining a personal advantage may shape the interpretation and depends on the specific situation (penalty decision for or against one's own team) [61]. However, when not personally involved, as in the setting of this study, the results suggest a general preference for matches to be decided by open play rather than by penalty kicks.

Indeed, the analysis of Consensus revealed systematic divergence across Stakeholder Groups, with more than half of the handball incidents receiving significantly different decision distributions. Moreover, some incidents resulted in an even split within Stakeholder Groups (e.g., 50% of players deeming an incident punishable and 50% not punishable). These findings highlight a core issue: even when referees make decisions that align with UEFA's official guidelines, substantial disagreement persists among other stakeholders, even when they are not personally involved. This suggests that the underlying problem lies not only in decision accuracy but also in the inherent subjectivity of the handball law. Given that widespread disagreement is likely to persist regardless of how well referees apply the current guidelines, the development of clearer, more objective criteria for handball incidents may help reduce stakeholder dissatisfaction. These new guidelines could visualise arm positions for specific game situations that are allowed or disallowed, substantiating the current generic wording of the law. This would also facilitate broader communication and training on the handball guidelines for coaches and players. In addition, a stronger exchange of expectations between Stakeholder Groups could be beneficial. The findings indicate that even for Handball Categories with high Accuracy among referees (such as Arm supporting body and Blocked shot), coaches and players systematically differed in their decision-making, namely in every single handball incident belonging to these categories. This suggests that greater educational efforts, combined with more mutual dialogue around expectations, could help bridge the gap between referees and practitioners and potentially reduce controversy. While improving Consensus off the pitch may not fully eliminate dissent during matches [38], establishing a shared understanding of the Laws is a necessary foundation for progress [39] which seems to be currently lacking.

The divergence in perspectives is clearly reflected in the key difference in Reasoning observed between Stakeholder Groups. Referees primarily reasoned their decisions on factors explicitly outlined in the Laws of the Game, particularly focusing on Naturalness. In contrast, coaches and players placed greater emphasis on contextual factors that are not formally considered in the Laws, such as Avoidability and Impact. This suggests a misalignment between the criteria stated in the Laws and applied by referees versus those valued by practitioners – reflecting the theoretical tension between technical accuracy and perceived adequacy [33,62]. While referees prioritised accuracy by focusing on the strict application of written rules (e.g., Naturalness), practitioners appeared to evaluate the adequacy of the decision based on the incident's consequences (e.g., Impact). Consequently, when a minor technical infringement results in a severe sanction like a penalty kick, the decision may be technically accurate but perceived as inadequate by those affected. Governing bodies should consider whether situational context should be explicitly incorporated into the Laws of the Game to allow referees more accurate and adequate decisions at the same time [52]. This is particularly relevant for Impact, as coaches and players appear to value the consequences of a handball incident more than referees and governing bodies do – whether a handball prevents a goal-scoring opportunity or if the ball would otherwise go out of the penalty area, whether it significantly changes the trajectory of the ball or if the handball is hardly noticeable and the ball reaches its destination anyway. The argument that a potentially match-deciding event such as a penalty kick should (only) be awarded if the handball offence is equally severe is a reasonable perspective that deserves further consideration in future revisions of

the handball law. It is important to note that this research was conducted before EURO 2024 and the controversy surrounding the quarter-final handball incident involving Spain and Germany. This ensures that the results were not influenced by the public debate in Germany that followed the tournament. However, the observed differences in Reasoning offer an empirical explanation for the mechanics of this controversy. While the on-field decision was likely based on the referee's assessment of Naturalness, the massive dissent from practitioners and the public was arguably driven by the high Impact of the incident (preventing a potential goal). The fact that UEFA later clarified the incident as punishable highlights the inherent ambiguity of the Naturalness criterion. This real-world example not only illustrates the clash between technical assessments and perceived adequacy but also underscores the critical need for clearer, more objective criteria to prevent such discrepancies in interpretation.

### Limitations and future directions

While this research is the first to provide valuable insights into the interpretation of handball incidents by referees, coaches, and players, some limitations must be acknowledged.

First, the studies relied on video-based assessments rather than real-time on-field decisions. While the virtual setting allowed for controlled conditions, it does not fully replicate the dynamic, high-pressure environment of real matches [47]. Regarding visual perception, prior research suggests that slow motion inflates the perceived duration of sporting actions, which can influence the perception of intent [24]. However, professional referees' perceptions of intent remain stable under slow motion [26], whereas participants without referee training might be more susceptible to changes in playback speed [25]. Interestingly, coaches and players in the present study exhibited significantly lower Strictness than referees and the UEFA baseline. Furthermore, no statistically significant differences in the Reasoning Category Intent were observed between stakeholders, despite the use of slow-motion footage. This could indicate that these expert practitioners share the referees' ability to maintain stable perceptions of intent. However, as this study's design does not allow drawing reliable conclusions on this specific effect, future research should explicitly investigate how coaches and players perceive real-time versus slow-motion incidents.

Closely related to the video setting is the presence of visible contextual information, such as match score and time. While participants were instructed to judge incidents strictly according to the Laws of the Game, previous research indicates that referees consciously or unconsciously use such contextual cues for game management [35,63,64], adjusting their decision-making thresholds individually [36]. Although this study focused on technical accuracy, it cannot be ruled out that the visible context influenced the 'adequacy' assessment of some participants, leading them to withhold a penalty decision in ambiguous situations [34]. Future studies could systematically manipulate these contextual constraints to isolate their specific impact on handball interpretations.

Second, although the study included referees, coaches, and players from German men's professional football, the findings may not be fully generalisable to other leagues, levels of play, or women's football. Differences in refereeing education or examples of prominent handball decisions could influence how handball incidents are interpreted in other contexts. Future studies should explore whether similar discrepancies exist across different football cultures, levels of competition, or gender.

Third, the sample sizes for coaches ($n=31$) and players ($n=46$) were smaller than the sample of referees ($n=154$). While significant differences were still observed, larger samples of coaches and players could provide an even more comprehensive understanding of how these stakeholders interpret handball incidents. However, given that all participants were drawn from elite football, the sample size can still be considered substantial within this specialised population. Moreover, the number of handball incidents was reduced in Study 2 to shorten the survey duration. A higher number of scenes per Handball Category could make findings regarding the categories more reliable. Future research could focus on specific Handball Categories like Blocked cross to deepen the understanding of the controversy in such sub-areas. In this context, employing eye-tracking technology would allow to objectively analyse whether stakeholders attend to the relevant diagnostic cues in these complex situations [65,66].

Fourth, while the study examined differences in Reasoning across Stakeholder Groups, the predefined reasoning options in Study 2 were derived from recurring responses provided by referees in Study 1. This approach ensured comparability but may have influenced the chosen explanations by coaches and players towards the predefined options, despite the inclusion of a free-text option. Furthermore, Reasoning was analysed at a global level, comprising all Handball Categories within the study. Future research could use more open-ended qualitative methods to explore reasoning factors by other stakeholders and conduct analyses differentiated by category and decision (punishable vs. not punishable).

Nevertheless, the studies provide a strong foundation for understanding differences in handball interpretation among key football stakeholders and highlight areas for potential improvements in referee education and the formulation of clearer handball guidelines. Future studies could expand the scope to include interpretations from further football stakeholders like the media and fans.

## Conclusion and practical implications

The findings of this research have several implications for football's governing bodies and referee education. Referee training programmes should aim to further enhance consistency both across Referee Role Groups and between different Performance Levels. The observed discrepancies in Strictness between UEFA, referees, and practitioners suggest that governing bodies should reassess whether their interpretations align with the practical expectations of the game or if in certain situations a more lenient approach would increase acceptance – particularly given the significant influence that penalty kicks can have on match outcomes. The lower Accuracy observed among practitioners highlights fundamental differences in how handball incidents are interpreted. A deeper, systematic issue lies in the differing interpretations within Stakeholder Groups themselves. Even if referees were able to perfectly apply the governing bodies' guidelines, a substantial proportion of other stakeholders would still hold differing views. This underscores the need for clearer and more objective guidelines – especially for more ambiguous Handball Categories – to reduce inconsistency and controversy. These should specify allowed and disallowed arm positions during certain match situations. Incorporating aspects of situational context, especially the Impact of handball incidents, into the Laws of the Game may help align refereeing decisions more closely with the expectations of players and coaches and therefore increase acceptance of handball decisions. The joint development of clear and objective criteria is essential, followed by transparent publication of the conventions. Achieving this will require close collaboration between stakeholders and effective communication.

## Supporting information

**S1 Dataset. Data analysed in Study 1.**
(CSV)

**S2 Dataset. Data analysed in Study 2.**
(CSV)

**S3 Dataset. Metadata on the 30 video scenes.**
(CSV)

**S4 Table. Demographic information about participants.**
(XLSX)

## Acknowledgments

The authors thank all referees, coaches, and players who contributed their time and expertise to this research. Further thanks are due to the German Football Association (DFB), the Association of German Football Coaches (BDFL), and the participating clubs for supporting this research by facilitating contact with their athletes and members.

## Author contributions

**Conceptualization:** Tobias Bauch, Daniel Leyhr, David Schmidt, Daniel Brinkmann, Oliver Höner.

**Data curation:** Tobias Bauch, David Schmidt.

**Formal analysis:** Tobias Bauch.

**Investigation:** Tobias Bauch.

**Methodology:** Tobias Bauch, Daniel Leyhr, Oliver Höner.

**Project administration:** Tobias Bauch.

**Resources:** Daniel Brinkmann.

**Software:** Tobias Bauch.

**Supervision:** Oliver Höner.

**Validation:** Daniel Leyhr.

**Visualization:** Tobias Bauch.

**Writing – original draft:** Tobias Bauch.

**Writing – review & editing:** Daniel Leyhr, Oliver Höner.

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
