## [Decision Letter · Decision Letter 0]

25 Nov 2025

PONE-D-25-48492Penalty kick or not? Differences in the interpretation of handball incidents in professional association footballPLOS ONE?

Dear Dr. Bauch,

We look forward to receiving your revised manuscript.

Kind regards,

Job Fransen

Academic Editor

PLOS ONE

Journal Requirements:

“The authors have declared that no competing interests exist.

Some of the authors are employed by the German Football Association (DFB). However, this affiliation did not influence the research process or outcomes. The study was designed and conducted independently, and none of the potential results would have conferred specific advantages or disadvantages to the authors or the DFB.”

Reviewers' comments:

Reviewer's Responses to Questions

**Comments to the Author**

1. Is the manuscript technically sound, and do the data support the conclusions?

Reviewer #1: Partly

Reviewer #2: Yes

2. Has the statistical analysis been performed appropriately and rigorously?

Reviewer #1: Yes

Reviewer #2: Yes

3. Have the authors made all data underlying the findings in their manuscript fully available?

Reviewer #1: Yes

Reviewer #2: Yes

4. Is the manuscript presented in an intelligible fashion and written in standard English?

Reviewer #1: Yes

Reviewer #2: Yes

Reviewer #1: There seem to be some errors in the data, specifically in the Consensus data. The results and discussion section regarding the consensus could therefore not be assessed properly. Specifically:

Table 4: seem to have some errors in it, please check

• Number of coaches is 31, so max n should be 31, right? Scene 7 and 11 have n 34 and 32, that seem more than the total number of coaches. Is this an error?

• Scene 2 and 19 the percentages of coaches and players seem to be mixed up. Coaches have n 2 and 1 but percentages 0 and 0, while players have n 0 and 0 and percentage 4.4 and 2.2 (which is impossible with n =0). Other percentages and n also don’t seem to add up, so please check whole table! -> Also please check results section based on this, e.g. line 420-421 seem incorrect with data presented in table 4

Additionally, the discussion is oriented much towards social relevance and misses sometimes proper scientific substantiation. It seems the aim of the authors is to change the Laws of the Game, while it should be scientific relevance.

Reviewer #2: The authors conducted a very interesting study exploring referees, players, and coaches' perspectives of handball incidents in football. This was a very interesting study and I enjoyed reading it. Please see some comments below that hopefully help improve the manuscript.

L65-78 (and broader introduction/discussion sections): This is a really interesting discussion. You could consider drawing on research in officiating other sports (e.g. Australian rules football, Rugby, handball [pun unintended!]) to discuss subjective and objective decision-making. Further, you could also consider concepts such as decision-making thresholds, game management, and/or accuracy vs adequacy debate. These are common aspects in the officiating decision-making literature that would really strengthen this discussion.

L190: In your reporting, could you consider splitting the AREF group by separating assistant referees and VARs? This would provide an interesting comparison as their roles are very different, and the VARs would be experienced in watching video to make decisions, whether the assistant referees would not be.

L193: You could consider using the recently published officiating expert statement in your reporting, specifically performance level:

Webb, T., Hancock, D. J., Weston, M., Warner, S., Helsen, W. F., MacMahon, C., ... & Tingle, J. K. (2025). The future for sport officiating research: an expert statement. Managing Sport and Leisure, 1-10.

L210-212: Was the score and time of game included in the video clip? Either way, I would recommend discussing this in the discussion/limitations as to how the presence or absence of this information may have influenced the decision-making of the referees.

L218-219: Who conducted these interpretations - was it a member of the research team or an officiating expert? Or was this data made available to the research team?

L220-222: Could you explain this on-field review more? How were the videos occluded?

L223-225: Should this be reported in the results?

L226-227: Do you have the data on how many times participants replayed the videos?

L235-236: Was this rating for videos in Study 2 based on the referees’ rating in Study 1?

L486: This is a very interesting point. I would recommend the authors consider the implications around availability of VAR at different levels. For example, higher levels of officials have the luxury of using VAR that lower level officials do not. What recommendations would you have in your results for decision-making development of different levels of referees?

L494: You could also consider the following study as a discussion point for how officials apply decisions:

Russell, S., Renshaw, I., & Davids, K. (2019). How interacting constraints shape emergent decision-making of national-level football referees. Qualitative Research in Sport, Exercise and Health, 11(4), 573-588.

**Do you want your identity to be public for this peer review?** For information about this choice, including consent withdrawal, please see our Privacy Policy

Reviewer #1: No

Reviewer #2: **Yes:** Aden Kittel

---

## [Author Response · Author response to Decision Letter 1]

29 Dec 2025

Dear Editors and Reviewers,

thank you for your feedback on the original submission. The comments were very helpful, and we think by addressing them we could further increase the article’s quality and relevance for the scientific community. Please find our responses to each point raised during the review process below:

→ All style requirements were checked. The files belonging to the supporting information section were re-uploaded with adapted file names.

• Please find our updated competing interests statement. Could you please change the online submission form on our behalf:

“The authors have declared that no competing interests exist.

Some of the authors are employed by the German Football Association (DFB). However, this affiliation did not influence the research process or outcomes. The study was designed and conducted independently, and none of the potential results would have conferred specific advantages or disadvantages to the authors or the DFB. This affiliation does not alter adherence to PLOS ONE policies on sharing data and materials.”

• Please include your full ethics statement in the ‘Methods’ section of your manuscript file. In your statement, please include the full name of the IRB or ethics committee who approved or waived your study, as well as whether or not you obtained informed written or verbal consent. If consent was waived for your study, please include this information in your statement as well.

→ This information was included in the Methods section.

Reviewer #1:

• Errors in Table 4:

→ Indeed, two columns of the table were accidentally switched during the formatting of the document. We apologise for the mistake and thank you very much for pointing it out. The columns were reordered and all data and references in the text to it were checked.

• Scientific substantiation of the Discussion section:

→ The Introduction and Discussion sections were extended to cover additional aspects relevant to the scientific community.

Reviewer #2:

• L65-78 (and broader introduction/discussion sections): […] drawing on research in officiating other sports […] concepts such as decision-making thresholds, game management, and/or accuracy vs adequacy debate. […]

→ These topics were added to the introduction and discussion in the revised manuscript.

• L190: In your reporting, could you consider splitting the AREF group by separating assistant referees and VARs? This would provide an interesting comparison as their roles are very different, and the VARs would be experienced in watching video to make decisions, whether the assistant referees would not be.

→ This would be an interesting comparison indeed. Unfortunately, the number of specialised VARs was too small (n=8) to analyse them as a separate group. Therefore, these video assistants were included in the AREF group.

• L193: You could consider using the recently published officiating expert statement in your re-porting, specifically performance level:

Webb, T., Hancock, D. J., Weston, M., Warner, S., Helsen, W. F., MacMahon, C., ... & Tingle, J. K. (2025). The future for sport officiating research: an expert statement. Managing Sport and Leisure, 1-10.

→ This was considered in the methods section and an additional table with demographic in-formation was added in the supporting information section.

• L210-212: Was the score and time of game included in the video clip? Either way, I would recommend discussing this in the discussion/limitations as to how the presence or absence of this information may have influenced the decision-making of the referees.

→ Yes, those were visible in the video clips. This aspect is now addressed in the limitations section.

• L218-219: Who conducted these interpretations - was it a member of the research team or an officiating expert? Or was this data made available to the research team?

→ The interpretations were done by the UEFA Referees Committee and available to the re-search team. This was clarified in the revised manuscript.

• L220-222: Could you explain this on-field review more? How were the videos occluded?

→ This was clarified in the revised manuscript.

• L223-225: Should this be reported in the results?

→ We believe this information should remain (only) in the Methods section for better clarity by addressing concerns of a potential bias due to the visibility of OFRs directly.

• L226-227: Do you have the data on how many times participants replayed the videos?

→ No, this information could unfortunately not be recorded in the technical environment we used.

• L235-236: Was this rating for videos in Study 2 based on the referees’ rating in Study 1?

→ We clarified in the manuscript that the difficulty rating of 4.1 ± 2.6 refers to the data collected in Study 2.

• L486: This is a very interesting point. I would recommend the authors consider the implications around availability of VAR at different levels. For example, higher levels of officials have the luxury of using VAR that lower level officials do not. What recommendations would you have in your results for decision-making development of different levels of referees?

→ We considered this point thoroughly. However, we do not believe that the availability of the VAR has a direct influence on the results of our specific research question regarding handball interpretation. We added this aspect and our thoughts to the discussion.

• L494: You could also consider the following study as a discussion point for how officials apply decisions:

Russell, S., Renshaw, I., & Davids, K. (2019). How interacting constraints shape emergent decision-making of national-level football referees. Qualitative Research in Sport, Exercise and Health, 11(4), 573-588.

→ This study is referenced now in the expanded introduction section.

We hope that the uploaded revision of the manuscript now fully meets PLOS ONE’s publication criteria and are looking forward to your response.

Yours sincerely,

Tobias Bauch

---

## [Decision Letter · Decision Letter 1]

13 Jan 2026

Penalty kick or not? Differences in the interpretation of handball incidents in professional association football

PONE-D-25-48492R1

Dear Dr. Bauch,

We’re pleased to inform you that your manuscript has been judged scientifically suitable for publication and will be formally accepted for publication once it meets all outstanding technical requirements.

Kind regards,

Ender Senel, PhD

Academic Editor

PLOS One

Reviewers' comments:

Reviewer's Responses to Questions

**Comments to the Author**

Reviewer #1: All comments have been addressed

Reviewer #2: All comments have been addressed

2. Is the manuscript technically sound, and do the data support the conclusions?

Reviewer #1: (No Response)

Reviewer #2: Yes

3. Has the statistical analysis been performed appropriately and rigorously?

Reviewer #1: (No Response)

Reviewer #2: Yes

4. Have the authors made all data underlying the findings in their manuscript fully available?

Reviewer #1: (No Response)

Reviewer #2: Yes

5. Is the manuscript presented in an intelligible fashion and written in standard English?

Reviewer #1: (No Response)

Reviewer #2: Yes

Reviewer #1: (No Response)

Reviewer #2: Well done on your revisions. I recommend publication. Best of luck for your future research in this area.

**Do you want your identity to be public for this peer review?** For information about this choice, including consent withdrawal, please see our Privacy Policy

Reviewer #1: No

Reviewer #2: **Yes:** Aden Kittel

---

## [Editor Report · Acceptance letter]

PONE-D-25-48492R1

PLOS One

Dear Dr. Bauch,

I'm pleased to inform you that your manuscript has been deemed suitable for publication in PLOS One. Congratulations! Your manuscript is now being handed over to our production team.

Kind regards,

on behalf of

Dr. Ender Senel

Academic Editor

PLOS One